# Analysis of Ana/Dfs70 Pattern in a Large Cohort of Autoimmune/Autoinflammatory Diseases Compared with First Degree Relatives and Healthy Controls Evaluated from Colombia

**DOI:** 10.3390/diagnostics12092181

**Published:** 2022-09-09

**Authors:** Consuelo Romero-Sánchez, Omar-Javier Calixto, Veronica Romero-Alvarez, Alejandra Vargas-Martin, Luis Castro, Julio Amador, Daniela Marín-Acevedo, Mónica Acevedo-Godoy, Diana Rincón-Riaño, Juan Manuel Bello-Gualtero

**Affiliations:** 1Rheumatology and Immunology Department, Hospital Militar Central, Bogota 110111, Colombia; 2Clinical Immunology Group, Hospital Militar Central, School of Medicine, Universidad Militar Nueva Granada, Bogota 110111, Colombia; 3Cellular and Immunology Group/InmuBo, Universidad El Bosque, Bogota 110111, Colombia; 4Faculty of Basic Science, Universidad Colegio Mayor de Cundinamarca, Bogota 111311, Colombia; 5Hospital Militar Central, Dermatology Department, Universidad Militar Nueva Granada, Bogota 110111, Colombia; 6Immunology Laboratory, Hospital Militar Central, Bogota 110111, Colombia

**Keywords:** antinuclear antibodies, ANA/DFS70, healthy individuals, systemic autoimmune/autoinflammatory rheumatic diseases

## Abstract

Background: The presence of Antinuclear antibodies/Dense Fine Speckled 70 (ANA/DFS70) has been proposed as a negative biomarker in the process of exclusion of systemic autoimmune/autoinflammatory rheumatic diseases (SARD). The purpose was to evaluate and characterize ANA/DFS70 patients in a large Colombian population with SARD; rheumatoid arthritis (RA), Psoriasis (PsO), Undifferentiated connective tissue disease (UCTD), first-degree relatives of (FDR), and healthy controls (HC). Methods: ANA determination was performed using indirect immunofluorescence. Samples with positive dense fine granular staining in the nucleoplasm of the interphase cell (AC2) fluorescence were confirmed with CytoBead/ANA and ANA/modified (Knocked out for the PSPI1 gen). Results: 530 mestizo Colombian participants were included. ANA/DFS70 antibody positivity in the whole group was 2.3%, and 0.8% in SARD; no RA patients were positive. ANA/DFS70 positives in UCTD were three women; the average time of evolution of the disease was 9.4 years. The most frequent clinical findings were arthralgias, non-erosive arthritis, and Raynaud’s phenomenon. The PsO positive was a woman with C-reactive protein (CRP) positivity and a negative erythrocyte sedimentation rate (ESR) without any other positive autoantibody or extracutaneous manifestation. FDR and HC positives were 7/8 women. All were negative for other autoantibodies. Conclusions: ANA/DFS70 autoantibodies were present in Colombian patients with SARD at a shallow frequency, they were more prevalent in healthy individuals.

## 1. Introduction

Autoantibodies are critical elements in diagnosing systemic autoimmune diseases [1]. Antinuclear antibodies (ANA) have been a matter of concern for both clinicians and patients, given the possibility of having or being at risk of developing systemic autoimmune rheumatic diseases (SARD), even more, if there is a first-degree relative affected [2].

Antinuclear antibodies/Dense Fine Speckled 70 (ANA/DFS70) antibodies are a new dense-appearing fine speckled pattern at the nuclear level in immunofluorescence (IIF) in Hep-2 cells [3]. These autoantibodies target a 70 kDa protein also known as LEDGF (Lens Epithelium-Derived Growth Factor) product of the PSIP1 gene [4]. ANA/DFS70 antibodies presence has been evidenced in different diseases, such as atopic diseases, alopecia areata, ocular diseases, chronic fatigue syndrome, arthralgia, fibromyalgia, interstitial cystitis, prostate cancer, Bechet’s disease, and autoimmune thyroiditis [5,6,7].

Interestingly, ANA/DFS70 positivity has been reported in up to 10.7–17.4% of apparently healthy individuals in a wide range of titers [8,9]. ANA/DFS70 presence has been proposed as essential in the exclusion process of SARD as a negative biomarker [10,11,12,13]. However, this is still controversial; meta-analysis studies consider high heterogeneity in sensitivity and specificity in SARD exclusion [14]. Nevertheless, a recent meta-analysis including the detection of ANA/DFS70 antibodies, a chemiluminescence assay, enzyme-linked immunoassay (ELISA) and western blot with the Hep-2 substrate reported a sensitivity of 19% (95% confidence interval (CI) 12–18%) and specificity of 93% (95% CI 88–96%) for the exclusion of SARD in participants with a positive ANA result [15].

Considering the cost related to clinical follow-up it has been proposed that the use of the ANA/DFS70 test could avoid unnecessary analyses and consultations in the health system through a cost-effective analysis [16].

In Colombia, ANA/DFS70 has been previously described; a comparison between systemic lupus erythematosus (SLE) and healthy controls (HC), evidenced the presence of a higher frequency of positive ANA/DFS70 in HC at 33.3% compared to 12.5% in SLE (*p* = 0.005) [17]. ANA/DFS70 was measured in patients with rheumatoid arthritis (RA) and compared to first-degree relatives (FDR), and HC, identifying ANA/DFS70 in 1.7% of FDR, and 2.5% in HC; no RA were positive. Likewise, there was an association with a standard erythrocyte sedimentation rate (ESR) (*p* = 0.032), negative rheumatoid factor (RF) (*p* = 0.044), and absence of painful joint count (*p* = 0.039) [18]. In a complementary manner, an additional analysis was performed (unpublished data), evaluating ANA/DFS70 in patients with undifferentiated connective tissue disease (UCTD) with evidence of a positive result in 11.9%. Positive patients were older than 50 years, had a disease duration greater than 5 years, and had a predominance of articular and dry symptoms.

Therefore, the purpose of this study was to evaluate and characterize ANA and ANA/DFS70 patients’ positivity and the autoantibody profile in a large Colombian population with SARD: RA, Psoriasis (PsO), as well as UCTD, FDR, and HC.

## 2. Materials and Methods

### 2.1. Participants

#### 2.1.1. Rheumatoid Arthritis Patients and First-Degree Relatives of Rheumatoid Arthritis

The inclusion criteria for patients with RA were patients between 18 and 65 years old who met the American College of Rheumatology (ACR)/(European League Against Rheumatism (EULAR) 2010 classification criteria [19] and who had conventional treatment, and FDR of consanguinity defined according to the EULAR recommendations [20]. Attended by the Rheumatology and Immunology Service of the Hospital Militar Central and the Fundación Instituto de Reumatología, Fernando Chalem from Bogotá, Colombia, the clinimetric evaluation included joint count, Visual Analog Scale for Pain (VAS Pain) [21], Disease Activity Score-28 (DAS28) [22], Simplified Disease Activity Index (SDAI) [23], Routine Assessment of Patient Index Data 3 (RAPID3) [24], and the Modified Health Assessment Questionnaire (MHAQ) [25] performed by rheumatologists. In the family group, the presence of painful and swollen joints was evaluated as joint clinical variables, and systemic disease was ruled out.

#### 2.1.2. Undifferentiated Connective Tissue Disease

The inclusion criteria for patients with UCTD were patients between 18 and 65 years old who met the qualifying criteria [26], with more than a year of the evolution of the disease. They were attended to by the Rheumatology and Immunology Service of the Hospital Militar Central from Bogotá, Colombia.

#### 2.1.3. Psoriasis

The inclusion criteria for patients with PsO were patients between 18 and 65 years old who met the updated criteria of the 2012 Colombian Consensus of Psoriasis and the provision of informed consent [27]. For each patient, we used the Psoriasis Area and Severity Index (PASI) and Dermatology Life Quality Index (DLQI) to determine the clinical activity and impact on quality of life [28,29]. They were attended to by the Dermatology Service of the Hospital Militar Central from Bogotá, Colombia.

#### 2.1.4. Healthy Control

The HC group consisted of people between 18 and 65 years old who lived and worked similarly to the patients included. The exclusion criteria were having a blood relationship with patients with an autoimmune or autoinflammatory disease, infectious diseases, neoplasms, diabetes, antibiotic treatment, pregnancy, or lactation. They were invited to participate without economic incentives and were from Bogotá, Colombia.

### 2.2. Laboratory Analyses

All individuals’ levels of CRP were assessed with high sensitivity chemiluminescence (Immulite 1000, Siemmens^®^ Erlanhen, Germany), and ESR by fluorometry (Test 1 THL Ali FAX^®^ Bogota, Colombia). The determination of anti-citrullinated peptide (anti-CCP) antibodies was by an enzymatic method (Quanta lite^®^ CCP 3.1 IgG/IgA, INOVA Diagnostics, Santiago, Chile) or turbidimetry by Spinreact^®^ Barcelona, Spain.

For the determination of autoantibodies against extractable nuclear antigens (ENAS), ELISAs for anti-SS/A (Seraquest/QI01250^®^, anti-SS/B Seraquest/QI01260^®^, anti-RNP Seraquest/QI01240^®^ and anti-Sm Seraquest/QI01230^®^) and anti-Deoxyribonucleic Acid double-stranded (DNAds) antibodies (Quanta lite^®^ HA dsDNA IgG, INOVA Diagnostics^®^) were used. Assays were used for Anti-cardiolipin IgG/IgM/IgA antibody determinations (Generic Assay GA4016^®^ and Quanta lite ACA IgA, INOVA Diagnostics^®^, respectively) and anti-Beta 2 glycoprotein IgG/IgM antibodies (Generic Assay GA4041^®^).

#### ANA and ANA/DFS70 Measurement

ANA determination was performed using the IIF technique; the serum was reacted on the slides with the substrate Hep-2 1103 and Hep-2-DFS70 ref 1108, the Autoantibody test System IMCO Diagnostics^®^, knocked out the PC4 and SFRS1 Interacting Protein 1 (PSIP1) gene, which prevents binding sites for the 70 kDa protein recognized by these autoantibodies. These modified cells can recognize all other autoantibodies except DFS70. Therefore, the samples with positive AC-2 will be considered positive over the modified cells when the fluorescent reaction disappears.

An initial dilution in saline phosphate buffer of 1/80 was made up to the final titer, with those samples with fluorescence being positive with a pattern with dense, fine granular staining in the nucleoplasm of the interphase cell (AC-2), typically excluding the nucleoli with bright staining of the chromosomes in the mitotic cell phase. Each test was performed with their respective positive and negative controls. The confirmatory tests for ANA/DFS70 (Autoantibody test System Imco Diagnostics^®^ ref 1108), and CytoBead ANA-Generic Assay ref 8260^®^ were tested in all samples. For proper conservation, the ANA kits were stored at a temperature of 2 to 8 °C, according to the manufacturer’s recommendations. The slides were read by two experts independently, one of them being the gold standard with more than 25 years of experience in IIF reading. Microscope readings Mi5 Lumin Epifluorescence.

### 2.3. Statistical Analyses

A descriptive statistical analysis of the categorical variables in percentages and quantitative variables measured by central tendency according to the characteristics of said variables was performed. A comparison between groups was made using the Chi-square test or Fisher’s exact test for categorical variables based on the expected values of the contingency table. In the case of continuous variables normality of the distribution was tested using the Kolmogorov–Smirnov test, and then the nonparametric Kruskal–Wallis H test was performed. Therefore, an additional bivariate comparison was performed between non-ANA/DFS70 positive and ANA/DFS70 positive groups. The analyses were performed using the statistical program SPSS 26. *p*-values were considered statistically significant with a result of <0.05.

## 3. Results

A total population of 530 mestizo Colombian participants was included. The distribution by diagnostic group was RA 19% (*n* = 98), PsO 11.3% (*n* = 58), UCTD 8.9% (*n* = 47), FDR 25.8% (*n* = 134), and HC 35% (*n* = 181). The population characteristics are presently based on ANA positivity in Table 1; ANA negative (64.2%), non-ANA/DFS70 positivity (33.6%), and ANA/DFS70 2.3%.

The ANA negative group (*n* = 340) was 73% women and had a median age of 43 (32.5–52) years, 27.3% were former smokers and 7.6% were current smokers, they had a positive ESR of 20.2% and CRP of 57.2%; other autoantibodies were 32.6%.

Non-ANA/DFS70 positivity (*n* = 178) was 75.3% women and had a median age of 42 (32–53.3) years; 31.5% were former smokers and 9% were current smokers; they had high levels of ESR, 19.1% and CRP, 50%, in addition to other autoantibodies, 37.1%. The distribution among groups was 24.7% in HC, 23.1% in RA, FDR in 19.1%, UCTD in 18.5%, and PsO in 14.6%. The ANA/DFS70 antibody positivity was 2.3% (*n* = 12) among this group; SARD patients were 33.3% (three UCTD and PsO one); healthy participants were 66.7% (six HC and two FDR), and no RA patients were positive. Groups proportions among Non-ANA/DFS70 and ANA/DFS70 are presented in Figure 1. Additionally, IFI for ANA examples are presented in Figure 2.

First degree relatives (FDR), healthy controls (HC), Psoriasis (PsO), Rheumatoid arthritis (RA), and Undifferentiated connective tissue diseases (UCTD).

The clinical and serological variables of individual cases of ANA/DFS70 positive are presented in Table 2. Three UCTD patients tested positive for ANA/DFS70 antibodies. All were women and the average time of evolution of the disease was 9.4 years (±2.9 years). Only 1/3 of patients had a high titer of ANA/DSF70 antibodies (AC-2) (1/640). In this group of individuals, the positive ENA was anti-Sm 1/3 and anti-RNP 1/3, the positive anti-dsDNA was 2/3, and 1/3 was positive for anti-cardiolipin IgG and anti-IgG Beta 2 glycoprotein; the FR and the anti-CCP antibody were negative, the acute phase reactant as CRP and the ESR fell in the normal ranges. The most frequent clinical findings of UCTD with a positive test for ANA/DFS70 antibodies were arthralgias (*n* = 3), non-erosive arthritis (*n* = 2), Raynaud’s phenomenon (*n* = 1), xerostomia (*n* = 1), and xerophthalmia (*n* = 1). All these patients have more than 5 years of evolution as UCTD.

The PsO patient with ANA/DFS70 positivity was a woman and had CRP elevated and ESR in normal ranges, without any other positive autoantibody, she had no extracutaneous manifestation, she was under biological treatment with anti-Tumor Necrosis Factor therapy with mild disease PASI: 3 points and DLQI: 0 points.

FDR and HC with ANA/DFS70 positivity were 7/8 women, all were negative for other autoantibodies measured (RF, anti-CCP, and anti-DNA ds), 2/8 were ESR abnormal and 7/10 were CRP abnormal, 2/8 had a smoking history.

On the other hand, when comparing ANA negative among groups there were statistically significant differences in frequencies of UCTD (*p* < 0.001), FDR (*p* = 0.024), and HC (*p* = 0.001), as well as RF positivity (*p* = 0.049). Additionally, a bivariate analysis between the Non-ANA/DFS70 positivity and ANA/DFS70 positivity group identified a significant difference in RA diagnosis frequency (*p* = 0.049).

Finally, none of the ANAS/DFS70 positive participants) in the three years of follow-up developed any definitive autoimmune disease.

## 4. Discussion

ANA/DFS70 antibodies are a new pattern (AC-2) that produces a dense-appearing fine speckled pattern at the nuclear level by IIF on HEp-2 cell substrate [3]. ANA/DFS70 autoantibody presence in the literature has been reported in up to 17.4% of apparently healthy individuals [8,9], and due to ANA/DFS70’s low frequency in SARD, it has been proposed as a biomarker for rule-out SARD in patients without autoimmune/autoinflammatory clinical presentation [10,11,12,13].

In patients with connective tissue diseases (RA, UCTD, systemic sclerosis, spondylarthritis, psoriatic arthritis, and familial Mediterranean fever), the presence of ANA/DFS70 has been described in 4.6% (7.8% in patients with positive ANA in screening) in a population of Turkey [24]. A comparison with patients diagnosed with RA with evidence of ANA/DFS70 of 2.4% was also included in this population [30]. In a recent study conducted in Spain, the positivity of ANA/DFS70 in patients with RA was 1.7%, while the positivity rate was 12.9% only among patients positive for ANA [31]. In the face of the absence of positivity for ANA/DFS70 in our RA population, 23.1% were ANA positive

In a study carried out in Japan; it was found ANA/DFS70 frequency was 16.4% in HC. Compared with other populations of rheumatological autoimmune diseases related to antibodies, no statistically significant differences were found, except in dermatopolymyositis [32]; however, when the population of ANA/DFS70 positives without any other positive antibody was analyzed, it was higher in the healthy population than in all populations with rheumatological diseases (except RA). However, no association was identified with titer on IIF measurement. Results with higher frequencies were found than in our HC group.

Data from Japan’s population of health workers without diseases showed that 11% were ANA/DFS70 positive, representing 54% of the population with positive ANA tests. It was reported that positivity for ANA/DFS70 decreases with increasing age in a statistically significant manner [10].

In Colombia in 100 SLE patients, 102 SARD, and 200 HC, 56 subjects were suspected of having an autoimmune disease with ANA positive and negative anti dsDNA antibodies. ANA/DFS70 antibodies were positive in 1.8% of subjects with ANA positive/anti dsDNA negative, in 1% of SLE patients, 0.9% of patients with other SARD and only in 0.5% of HC [33]. A similar low frequency was found in our study with 2.3% but the frequency of ANA/DFS70 in other autoantibodies negative was just 0.5% (PsO patient).

In a cohort from Israel, a monospecific Anti-DFS70 analysis was performed with positivity of HC of 10.9%, and for patients with autoimmune rheumatic diseases of 1.9% [34]. A higher frequency was found than that reported in our study of 4.2% in healthy individuals (HC and FDR), and FDR could be participants with a higher risk for SARD.

Prevalence reports of ANAS/DFS70 in the literature, however, have been published in European, North American, and Asian populations [14,15,35,36,37], but few reports are presented in the Latin American population [7,38], especially in the Colombian population [17,18].

For example, in a cohort of Mexican patients, Anti/DFS70 was identified in 1.4% of dermatopolymyositis and polymyositis patients, RA in 4.3%, obesity in 6.6%, and 17.4% in the healthy population using different methods of diagnostics [38]. Data from Colombia [17,18], showed evidence of a higher frequency of positive Anti DFS70 in HC compared with SLE and RA patients. However, despite this difference, the frequency in HC is much lower than that reported in other latitudes, including the Mexican population [17,18,38], Colombia is not the exception; our population is composed of individuals with mixed ancestry generating ethnic blends. Since prehistoric times, Colombia has been subject to an intense genetic and cultural flow, mainly due to its geographical location, resulting in a high diversity of ethnic groups that inhabit the country and a notable heterogeneity between geographic regions increasing genetic variability [39]. Thus, adding a possible explanation for the low frequency of ANA/DFS70 in our results, as described with other biomarkers, such as anti-CCP in patients with early RA [40,41] and HLA-B27 in axial spondylarthritis [42].

In a complementary manner, an additional analysis was performed (accepted for publication but unpublished data) in patients with UCTD with evidence of a positive result in 9.4% that did not differentiate into a SARD in a five-year follow-up. In another study in 91 UCTD patients, ANA/DFS70 was present in 13.2% helpful in patients with a stable disease without progression to another autoimmune disease [43].

As a limitation, the present study design does not inform causality, so it is worth monitoring the levels of ANA and ANA/DFS70 before and after treatment to evaluate its behavior over time. We included a large population from different regions of Colombia as a referral center, but a great heterogeneity must be considered during data interpretation as the main objective was to assess the frequency of ANA/DFS70 which varied in clinical and laboratory characteristics, with differences in frequencies among SARD and HC, otherwise, all participants were evaluated by expert rheumatologists and dermatologists applying adequate validated classification criteria and health status confirmed by strict exclusion criteria. The slides were independently evaluated by IIF experts. There is vast information regarding IIF performance in daily practice, but in our study, the evaluation and report of ANA were made by two independent experts and a further confirmatory technique was implemented.

These results support the evidence that ethnicity influences the frequency of these autoantibodies but not in the distribution by group. ANA/DFS70 positivity and autoantibody negativity described abroad as a negative predictive marker of SARD implies a clinical and paraclinical follow-up. Hence, the importance that the inclusion of this marker may have to rule out autoimmune disease is highlighted.

## 5. Conclusions

Despite the low frequency in general, it has been shown that ANA/DFS70 is more prevalent in healthy individuals than in patients with SARD, which was found in this group of individuals. ANAS/DFS70 autoantibodies were present in Colombian patients with SARD at a shallow frequency. Thus, patients with a positive result tend to have a mild or non-progressing phenotype of autoimmune or autoinflammatory diseases. This analysis reinforces evidence of ANA/DFS70 positivity described abroad as a negative marker of SARD.

## Figures and Tables

**Figure 1 diagnostics-12-02181-f001:**
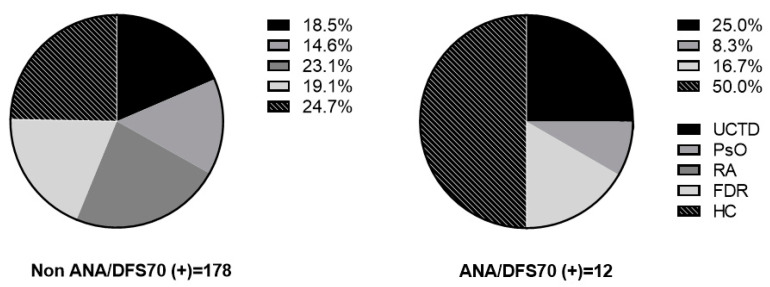
Frequency of participants based on positivity for non-ANA/DFS70 and ANA/DFS70. First degree relatives (FDR), healthy controls (HC), Psoriasis (PsO), Rheumatoid arthritis (RA), and Undifferentiated connective tissue diseases (UCTD).

**Figure 2 diagnostics-12-02181-f002:**
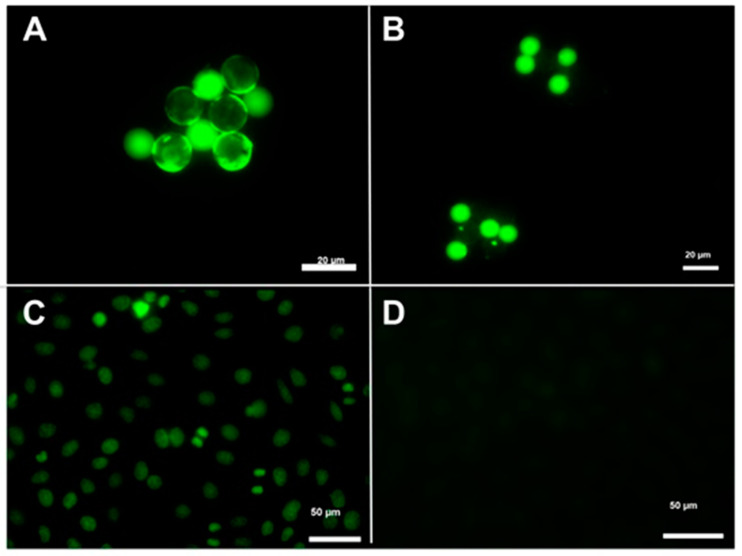
Indirect immunofluorescence for ANA. (**A**) Positive Cytobeads ANA for DFS70, positive antibodies in diluted patient serum react specifically with antigens on beads fixed onto slides/Cytobeads ANA for DFS70. Ring fluorescence of the antigen large, coated beads shows the presence of DFS70 and small bead small coated beads correspond to the internal control of the technique. 40×. (**B**) Positive internal control on coated small beads in an ANA/DFS70 negative patient sample. 40×. (**C**) ANA: Nuclear dense fine speckled AC-2 (Speckled pattern distributed throughout the interphase nucleus with characteristic heterogeneity in the size of Hep-2 cells. (**D**) ANA: Negative (AC-0). Absence of clear-cut staining in any given subcellular structure on Hep-2 cells. 40×. Nomenclature: International Consensus on Antinuclear Antibody (ANA) Patterns (ICAP). https://www.anapatterns.org/index.php (last accessed 23 June 2022).

**Table 1 diagnostics-12-02181-t001:** Clinical and demographical variables for included population.

	ANA (−) *n* (%)	Non-ANA/DFS70 (+) *n* (%)	ANA/DFS70 (+) *n* (%)	*p* Value
	*n* = 340	*n* = 178	*n* = 12	
UCTD *n* (%)	11 (3.2)	33 (18.5)	3 (25)	<0.001 *
PsO *n* (%)	33 (9.7)	26 (14.6)	1 (8.3)	0.207
RA *n* (%)	60 (17.6)	41 (23.1)	0 (0)	0.071
FDR *n* (%)	101(29.7)	34 (19.1)	2 (16.7)	0.024 *
HC *n* (%)	135 (39.7)	44 (24.7)	6 (50)	0.001 *
Gender				
Female *n* (%)	248 (73)	134 (75.3)	11 (91.7)	0.354
Age median (IQR)	43 (32.5–52)	42 (32–53.3)	37 (30.3–52.3)	0.073
Current smoker *n* (%)	26 (7.6)	16 (9)	1 (8.3)	0.395
Former smoker *n* (%)	99 (27.3)	56 (31.5)	2 (16.7)	0.067
ESR abnormal *n* (%)	69 (20.2)	34 (19.1)	2 (16.7)	0.948
CRP abnormal *n* (%)	195 (57.2)	89 (50)	8 (66.7)	0.237
Antibody profile				
RF + *n* (%)	34 (10)	29 (16.3)	0 (0)	0.049 *
ACPA + *n* (%)	64 (18.8)	31 (17.4)	0 (0)	0.231
SS/A + *n* (%)	0 (0)	6 (3.4)	0 (0)	0.145
SS/B + *n* (%)	1 (0.3)	6 (3.4)	0 (0)	0.297
Sm + *n* (%)	1 (0.3)	8 (4.5)	1 (8.3)	0.431
RNP + *n* (%)	1 (0.3)	10 (5.6)	1 (8.3)	0.222
dsDNA + *n* (%)	53 (15.5)	32 (18)	2 (16.7)	0.756
ACA IgG + *n* (%)	0 (0)	1 (0.6)	1 (8.3)	0.587
ACA IgM + *n* (%)	0 (0)	2 (1.1)	0 (0)	1
B2GPI IgG + *n* (%)	1 (0.3)	3 (1.7)	1 (8.3)	1
B2GPI IgM + *n* (%)	1 (0.3)	1 (0.6)	1 (8.3)	0.576

* Fisher exact test *p* < 0.05. Rheumatoid arthritis (RA), Psoriasis (PsO), Undifferentiated connective tissue diseases (UCTD), first degree relatives (FDR), healthy controls (HC), rheumatoid factor (RF), anti-citrullinated protein antibody (ACPA), Eritrosedimentation rate (ESR), C-reactive protein (CRP), Anti-cardiolipin IgG (ACA IgG), Anti-cardiolipin IgM (ACA IgM), anti-IgG Beta 2 glycoprotein (B2GPI IgG), anti-IgM Beta 2 glycoprotein (B2GPI IgM), double-stranded DNA (dsDNA), interquartile range (IQR).

**Table 2 diagnostics-12-02181-t002:** Clinical and demographical variables for ANAS/DFS70 patients *n* = 12.

	Age	Gender	ANA	ESR	CRP	RF	ACPA	SSA	SSB	Sm	RNP	dsDNA	ACA IgG	ACA IgM	B2GP IgG	B2GP IgM
UCTD	30	F	1/160	−	−	−	−	−	−	−	−	−	+	−	+	+
UCTD	61	F	1/320	−	−	−	−	−	−	+	+	+	−	−	−	−
UCTD	42	F	1/160	−	−	−	−	−	−	−	−	+	−	−	−	−
PsO	43	F	1/320	−	+	−	−	−	−	−	−	−	−	−	−	−
FDR	31	M	1/80	−	+	−	−	−	−	−	−	−	−	−	−	−
FDR	32	F	1/160	+	+	−	−	−	−	−	−	−	−	−	−	−
HC	50	F	1/80	−	+	−	−	−	−	−	−	−	−	−	−	−
HC	26	F	1/80	+	+	−	−	−	−	−	−	−	−	−	−	−
HC	27	F	1/160	−	−	−	−	−	−	−	−	−	−	−	−	−
HC	54	F	1/320	−	+	−	−	−	−	−	−	−	−	−	−	−
HC	32	F	1/160	−	+	−	−	−	−	−	−	−	−	−	−	−
HC	53	F	1/320	−	+	−	−	−	−	−	−	−	−	−	−	−

Psoriasis (PsO), Undifferentiated connective tissue diseases (UCTD), first degree relatives (FDR), healthy controls (HC), Antinuclear antibody ANA, rheumatoid factor (RF), anti-citrullinated protein antibody (ACPA), Eritrosedimentation rate (ESR), C-reactive protein (CRP), Anti-cardiolipin IgG (ACA IgG), Anti-cardiolipin IgM (ACA IgM), anti-IgG Beta 2 glycoprotein (B2GPI IgG), anti-IgM Beta 2 glycoprotein (B2GPI IgM), double-stranded DNA (dsDNA).

## Data Availability

Not applicable.

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
