# Peer review of "Analysis of Ana/Dfs70 Pattern in a Large Cohort of Autoimmune/Autoinflammatory Diseases Compared with First Degree Relatives and Healthy Controls Evaluated from Colombia"

_diagnostics, 2022, doi:10.3390/diagnostics12092181_

Round 1

Reviewer 1 Report

The manuscript written by Romero-Sánchez et al. describes the analysis of the presence of ANA/DFS70 autoantibodies connected with autoimmune/autoinflammatory diseases.

Below, there are some examples, which in my opinion would help to improve the quality of the manuscript.

1.     Abstract:

-        missing explanation of the abbreviation ANA/DFS70

-        in my opinion sentence needs language correction “Samples with fluorescence positive with the dense fine granular staining in the nucleoplasm of the interphase cell (AC2), were confirmed Cy-toBead/ANA and knocked out for the psip gene”

-        there is a typo in the final sentence: “Conclusions: AN-AS/DFS70 autoantibodies were present in Colombian patients with SARD at a shallow frequency, they were more prevalent in healthy individuals.”

2.     Introduction:

-        General: abbreviations should be explained at the first use (there are in the abstract)

-        For example: Line 40, line 48, line 50, there is unnecessary dash as a result of the text formatiing: “sys-temic”, “pros-tate”, “individ-uals” etc. The text is formatted carelessly.

-        Line 66: “identifying antibodies in 0%” in my opinion should be rewrited.

-        Line 81, text formatting: “ACR / EULAR 2010”

3.     Materials and methods

-        How many patients with certain disease were examined?

-        In the autoimmune diseases very important factor is the gender. Authors do not comment in any way the gender in this materials and methods paragraph.

-        Line 128: lack of abbreviation explanation for IIF

-        What was the diluent? “An initial dilution of 1/80 was made up to the final titer”

4.     Results

-        Formatting: “3.2. . Figures and Tables”

Discussion and analysis of the obtained results is comprehensive. I suggest precise text refinement in terms of language and formatting.

Author Response

Comments and Suggestions for Authors

Below, there are some examples, which in my opinion would help to improve the quality of the manuscript.

          Observation 1: Abstract:- missing explanation of the abbreviation ANA/DFS70

Response 1: Abbreviation was included

Observation 2: in my opinion sentence needs language correction “Samples with fluorescence positive with the dense fine granular staining in the nucleoplasm of the interphase cell (AC2), were confirmed Cy-toBead/ANA and knocked out for the psip gene”

Response 2: Sentence was changed for clarification.

Observation 3:  there is a typo in the final sentence: “Conclusions: AN-AS/DFS70 autoantibodies were present in Colombian patients with SARD at a shallow frequency, they were more prevalent in healthy individuals.”

Response 3:Typo was deleted .

Observation 4: Introduction: General: abbreviations should be explained at the first use (there are in the abstract) For example: Line 40, line 48, line 50, there is unnecessary dash as a result of the text formatiing: “sys-temic”, “pros-tate”, “individ-uals” etc. The text is formatted carelessly.

Response 4: Dashes were eliminated, the text format was adjusted and abbreviations were defined in first mention in the text

Observation 5: Line 66: “identifying antibodies in 0%” in my opinion should be rewrited.

             Response 5: The phrase was rewritten for clarification

Observation 5:Line 81, text formatting: “ACR / EULAR 2010”

Response 5:The phrase was rewritten for clarification

Observation 6: Materials and methods. How many patients with certain disease were examined?

Response 6: Total number were included with percentage for each population in lines 162-163

Observation 7: In the autoimmune diseases very, important factor is the gender. Authors     do not comment in any way the gender in this materials and methods paragraph.

Response 7: Thanks for your commentary, gender is significant for autoimmune diseases and this variable was measured. Nonetheless, in statistical analysis gender did not reach statistical difference significance as described in table 1. Then no adjustment for gender was performed in the further results

Observation 8: Line 128: lack of abbreviation explanation for IIF

Response 8: Abbreviation was defined

Observation 9: What was the diluent? “An initial dilution of 1/80 was made up to the final titer”

Response 9: The diluent is added in methods: saline phosphate buffer

Observation 10: Results-Formatting: “3.2. . Figures and Tables”

Response 10:  formatting was adjusted and subheading was removed

Observation 11: Discussion and analysis of the obtained results is comprehensive. I   suggest precise text refinement in terms of language and formatting.

               Response 11: Thanks for the observation, all the writing of the manuscript was revised. The English of the manuscript was revised in its entirety.  

Reviewer 2 Report

The topic of the article is very important because the discrimination between DFS and so-called homogeneous pattern might be a particularly challenging task for routine diagnostic laboratories and inaccurate interpretation may have significant consequence. Therefore, the presence of isolated anti-DFS70 antibodies could be used as a biomarker to exclude the diagnosis of ANA-associated autoimmune rheumatic diseases.

But the article is very flawed, written in very poor English and consequently poorly understandable and difficult to read:

1.    In the introduction, the field is not presented transparently, considering important literature references - the reference list does not cover the relevant literature adequately.

2.    Abbreviations used in article are inconsistent (PsO and/or PS, FDR and/or FR, ANA /DFS70 and/or ANAS /DFS70, HEP-2 and/or Hep-2, ANA (+) and/or non-ANA DFS70(+)) and very disturbing in reading and understanding. 

3.    The English grammar is poor and as some sentences are incomprehensible (lines: 59, 162, 164, 165, 178, 231, 249, 250 and many others). Because of that the article is poorly understandable and difficult to read.

4.    In the section Material and methods, the total number of patients enrolled in the study is not described.

5.    In the section Material and methods (line 119), it is wrong written “for determination of extractable nuclear antigens (ENAS)”, you detected antibodies against extractable nuclear antigens.

6.    In the section Statistical analyses, you have listed the following statistical tests: Chi-square test or Fisher's exact test, Mann Whitney T-test (better: Student t-test), or U-Mann Whitney test (better: Mann-Whitney U-test):

    1. It is not specified if both tests, the Chi-square test and Fisher's exact test, are used? If yes, it is not indicated Table 1 in which cases you used them. If you used only Fisher's exact test, as shown in Table 1, then you should describe only that test in the text.
    2. Did you test your data for normality (only age is shown as continuous?)? If so, which test did you use? Describe in the text.  
    3. t-test or Mann-Whitney U-test: these two tests are used to compare two independent groups. However, you have three independent groups in Table 1 (ANA (-), ANA (+), and ANA /DSF70 (+)). If I understand your table correctly, only age is expressed as continuous, so you need to use a test for comparing three independent groups. Please correct this in the text and describe it in Table 1.

 7.    Table 1:

a.    The table needs to be redesigned a bit to make it more transparent (trait groups visible) 

b.    Write down what data you are presenting in the table (examples: Age, median (IQR or 25th percentile-75th percentile), gender, female n (%). 

c.     I suggest providing a number and percentage as n (%) for all groups in the table because now it is not clear what the numbers represents.

    1. You must change SSa into SS/A nad SSb into SS/B
    2. Specify another autoantibody (………..) 

8.    You also write Hep-2 methods in line 56. It is wrong because the method is for example indirect immunofluorescence or chemiluminescence, but notHEp-2 cells which are substrate.

9.    In the section Results

    1. line 161 wrong diction is also healthy patients. You can use healthy parsons/individuals but not patients.
    2. Line 178 you wrote the acute phase reac-tants CRP and ESR…… ESR is not acute phase reactant
    3. The results are not clearly presented.

10. The section discussion must be more comprehensive.

Therefore, if the above shortcomings are not remediable, I do not support the publication of the article in Diagnostics.

Author Response

Comments and Suggestions for Authors

The topic of the article is very important because the discrimination between DFS and so-called homogeneous pattern might be a particularly challenging task for routine diagnostic laboratories and inaccurate interpretation may have significant consequence. Therefore, the presence of isolated anti-DFS70 antibodies could be used as a biomarker to exclude the diagnosis of ANA-associated autoimmune rheumatic diseases.

But the article is very flawed, written in very poor English and consequently poorly understandable and difficult to read:

             Observation 1. In the introduction, the field is not presented transparently, considering important literature references - the reference list does not cover the relevant literature adequately.

             Response 1: Thanks for the observation, the introduction was reviewed and adjusted.

            Observation 2: Abbreviations used in article are inconsistent (PsO and/or PS, FDR and/or FR, ANA /DFS70 and/or ANAS /DFS70, HEP-2 and/or Hep-2, ANA (+) and/or non-ANA DFS70(+)) and very disturbing in reading and understanding. 

         Response 2: Thanks for the observation. All abbreviations have been checked.

        Observation 3:The English grammar is poor and as some sentences are incomprehensible (lines: 59, 162, 164, 165, 178, 231, 249, 250 and many others). Because of that the article is poorly understandable and difficult to read.

        Response 3: Thanks for the observation, the manuscript was adjusted

               Observation 4:  In the section Material and methods, the total number of patients enrolled in the study is not described.

          Response 4: Total number were included with percentage for each population in lines 162-163

                    Observation 5: In the section Material and methods (line 119), it is wrong written “for determination of extractable nuclear antigens (ENAS)”, you detected antibodies against extractable nuclear antigens.

                    Response 5: thanks for the observation, abbreviation was corrected

           Observation 6: In the section Statistical analyses, you have listed the following statistical tests:  Chi-square test or Fisher's exact test, Mann Whitney T-test (better: Student t-test), or U-Mann Whitney test (better: Mann-Whitney U-test):

                   Response 6: the statistical tests names were corrected

                Observation 7: It is not specified if both tests, the Chi-square test and Fisher's exact test, are used? If yes, it is not indicated Table 1 in which cases you used them. If you used only Fisher's exact test, as shown in Table 1, then you should describe only that test in the text.

           Response 7: the statistical analysis was complemented explaining categorical and continuous variables tests performed

           Observation 8: Did you test your data for normality (only age is shown as continuous?)? If so, which test did you use? Describe in the text.  

           Response 8: the statistical analysis was complemented adding normality distribution test information

         Observation 9: t-test or Mann-Whitney U-test: these two tests are used to compare two independent groups. However, you have three independent groups in Table 1 (ANA (-), ANA (+), and ANA /DSF70 (+)). If I understand your table correctly, only age is expressed as continuous, so you need to use a test for comparing three independent groups. Please correct this in the text and describe it in Table 1.

   Response 9: the statistical analysis was corrected as more than 2 groups were compared the Kruskal-Wallis H test was performed. Additional explanation was included informing the additional comparison evaluated between non-ANA/DFS70 positive and ANA/DFS70 positive groups.

         Observation 10: Table 1: The table needs to be redesigned a bit to make it more transparent (trait groups visible) 

                Response 10: thanks for the recommendation the table was complemented with information for a clear interpretation

               Observation 11: Write down what data you are presenting in the table (examples: Age, median (IQR or 25th percentile-75th percentile), gender, female n (%). 

            Response 11: thanks for your commentary the table was completed with additional information for clarification

           Observation 12: I suggest providing a number and percentage as n (%) for all groups in the table because now it is not clear what the numbers represents.

Response 12: thanks for your commentary the table was completed with additional information for clarification

Observation 13: You must change SSa into SS/A and SSb into SS/B

Response 13: changed was made in the table 1, table 2, and methods section

      Observation 14: Specify another autoantibody (………..) 

Response 14: the meaning of another antibody positive refers to the presence of any antibody tested in the study as RF ACPA SS/A SS/B Sm RNP dsDNA ACA or B2GPI, otherwise, to avoid misinterpretation in the table this row was deleted

      Observation 15: You also write Hep-2 methods in line 56. It is wrong because the method is for example indirect immunofluorescence or chemiluminescence, but notHEp-2 cells which are substrate.

                Response 15: thanks for the observation the term was adjusted

                Observation 16: In the section Results: line 161 wrong diction is also healthy patients. You can use healthy parsons/individuals but not patients.

                Response 16: thanks for the observation the term was adjusted

    Observation 17: Line 178 you wrote the acute phase reactants CRP and ESR…… ESR is not acute phase reactant

                Response 17: thanks for the observation the term was adjusted

   Observation 18: The results are not clearly presented.

   Response 18: thanks for the observation the wording of the results was revised

                Observation 19:  The section discussion must be more comprehensive.

         Response 19: thanks for the observation the wording of the results was revised.  
